# Phylogenetic Investigations of Dengue 2019–2021 Outbreak in Guadeloupe and Martinique Caribbean Islands

**DOI:** 10.3390/pathogens12091182

**Published:** 2023-09-20

**Authors:** Margot Garcia--Van Smévoorde, Géraldine Piorkowski, Loic Emboulé, Georges Dos Santos, Cécile Loraux, Stéphanie Guyomard-Rabenirina, Marie-Odile Joannes, Laurence Fagour, Fatiha Najioullah, André Cabié, Xavier de Lamballerie, Anubis Vega-Rúa, Raymond Césaire, Elodie Calvez

**Affiliations:** 1Vector Control Research Laboratory, Transmission Reservoir and Pathogens Diversity Unit, Institut Pasteur de la Guadeloupe, Les Abymes 97139, Guadeloupe; mgarciavansmevoorde@pasteur-guadeloupe.fr; 2Unité des Virus Emergents (UVE), Aix-Marseille Univ-IRD 190-Inserm 1207, 13005 Marseille, France; geraldine.piorkowski@inserm.fr (G.P.); xavier.de-lamballerie@univ-amu.fr (X.d.L.); 3National Reference Center for Arboviruses, National Institute of Health and Medical Research (Inserm), 13005 Marseille, France; 4French Armed Forces Biomedical Research Institute (IRBA), 91220 Brétigny-sur-Orge, France; 5Molecular Genetics and Inherited Red Blood Cell Diseases Laboratory, University Hospitals of Guadeloupe, Pointe-à-Pitre 97159, Guadeloupe; loic.emboule@chu-guadeloupe.fr; 6Department of Virology, University Hospitals of Martinique, Fort-de-France, 97200 Martinique, France; georges.dos-santos@chu-martinique.fr (G.D.S.); laurence.fagour@chu-martinique.fr (L.F.); 7Pathogenesis and Controle of Chronic and Emerging Infections, French National Blood Service (EFS), National Institute of Health and Medical Research (Inserm), University of Montpellier, 34000 Montpellier, France; andre.cabie@chu-martinique.fr (A.C.); raymond.cesaire@chu-guadeloupe.fr (R.C.); 8University of Antilles, Pointe-à-Pitre 97110, Guadeloupe; 9Department of Virology, University Hospitals of Guadeloupe, Pointe-à-Pitre 97159, Guadeloupe; cecile.loraux@chu-guadeloupe.fr; 10Microbial Ecosystems Interaction Laboratory, Transmission Reservoir and Pathogens Diversity Unit, Institut Pasteur de la Guadeloupe, Les Abymes 97139, Guadeloupe; sguyomard@ch-tarbes-vic.fr; 11Department of Hematology Immunology Histocompatibility, University Hospitals of Guadeloupe, Pointe-à-Pitre 97159, Guadeloupe; marie-odile.joannes@chu-guadeloupe.fr; 12Department of Clinical Research and Innovation, University Hospitals of Martinique, Fort-de-France, 97200 Martinique, France; fatiha.najioullah@chu-martinique.fr; 13Department of Infectious and Tropical Diseases, University Hospitals of Martinique, Fort-de-France, 97200 Martinique, France

**Keywords:** dengue, outbreak, phylogeny, Guadeloupe, Martinique, Caribbean

## Abstract

Dengue fever has been a public health problem in the Caribbean region since 1981, when it first reappeared in Cuba. In 1989, it was reported in Martinique and Guadeloupe (two French islands 200 km apart); since then, DENV has caused several epidemics locally. In 2019–2021, DENV-1, DENV-2, and DENV-3 were detected. Serotype distribution was differentiated, with DENV-2 and DENV-3 predominating in Guadeloupe and Martinique, respectively. Complete genome sequencing was carried out on 32 specimens, and phylogenic analysis identified the circulation of genotype V for DENV-1, cosmopolitan genotype for DENV-2, and genotype III for DENV-3. However, two distinct circulating groups were identified for DENV-1 and DENV-3, suggesting independent introductions. Overall, despite the context of the COVID-19 pandemic and the associated travel restrictions, these results confirm the active circulation of DENV and specific epidemiological features on each of the two islands. Such differences may be linked to the founder effect of the various introduction events, and to local factors such as the population immunity and the transmission capacity of the vectors. Further genomic and epidemiological characterization of DENV strains remains essential to understand how dengue spreads in each specific geographical context and to prevent future epidemics.

## 1. Introduction

Dengue is a disease resulting from the infection of dengue virus (DENV), which reemerged worldwide at the end of the 20th century [1,2]. It is now the most common viral disease transmitted by arthropods in the world, and its incidence has increased more than tenfold over the last 20 years [3]. It is estimated that more than 390 million infections occur each year in the world, of which 96 million patients manifest symptoms. Dengue outbreaks are mainly observed in tropical and subtropical regions [2,4]. However, in recent years, autochthonous dengue cases have been reported in new areas such as in Europe and North America, as well as in Japan [5,6,7], and there is still no antiviral therapy or widely approved vaccine against this infection [8]. Furthermore, despite extensive vector control efforts, the global dengue outbreak distribution is rising linked to the intensification of trade and travel due to commerce and tourism [2,4,9,10].

DENVs are single-stranded, positive-sense RNA viruses (family *Flaviviridae*), and their genome encodes three structural and five nonstructural proteins [11]. There are four antigenically and genetically distinct serotypes (DENV-1, DENV-2, DENV-3, and DENV-4), which emerged from sylvatic cycles in Asia [12,13,14,15]. An infection with one DENV serotype confers a specific immunity against this serotype but no cross-protection against the other serotypes [9]. Serotypes are themselves subdivided into several genotypes mainly associated to the geographical origin of viral strains. For DENV-1 and DENV-3 serotypes, five genotypes numbered from genotypes I to V were reported. For DENV-2, six genotypes were recorded and named Asian I, Asian II, cosmopolitan, Asian/American, American, and Sylvatic. Lastly, for DENV-4, four genotypes were described (genotypes I to IV) [16].

In the Americas, dengue cases were clinically described for the first time in 1780 [17]. In many South American countries, since the reintroduction of the main vector *Ae. aegypti* in the 1970s, DENV circulation of the four DENV serotypes increased, and outbreaks were recorded every year [2]. In 2019, more than 3.1 million dengue cases were recorded in the American continent.

The Caribbean follows this increasing trend and is one of the most DENV-affected region in the Americas. After *Ae. aegypti* reintroduction in the 1970s [2], the first epidemic of dengue hemorrhagic fever (DHF) in the Caribbean occurred in Cuba in 1981 [18]. During the 1980s and 1990s, dengue spread throughout the Americas and Caribbean. 

In the French territories of America (FTA), dengue was described for the first time in 1977 in Martinique with the circulation of DENV-1. DENV-4 was then detected in 1982 in Guadeloupe and Martinique, followed by dengue outbreaks due to DENV-2 and DENV-3 serotypes in 1989 and 1999, respectively [19]. Since the 1990s, the dynamics of dengue epidemics on these islands intensified with the co-circulation of the four DENV serotypes, often with predominance of a serotype [20,21,22]. Between 1999 and 2010, the circulation patterns of DENV serotypes were similar between these islands. The reintroductions of DENV-3 in 1999 and DENV-4 in 2005 were simultaneous and caused major outbreaks in both islands [23,24]. Prominent circulation was observed of DENV-2 in 2007–2008, and of DENV-1 in 2009–2010. The DENV-1 epidemics were characterized by their magnitude and duration, with estimations of more than 40,000 symptomatic cases (>10% of the populations) in both islands [19]. However, since the 2013–2014 epidemic, the predominant DENV serotype involved in outbreaks has differed between the two islands [22]. Indeed, DENV-4 and DENV-2 co-circulated, but the main serotype was DENV-4 in Guadeloupe and DENV-2 in Martinique during the 2013–2014 outbreak [22].

During the last major outbreak in 2019–2021 in the French Caribbean islands, 23,690 and 33,120 clinical suspected cases of dengue fever were estimated in Guadeloupe and in Martinique, respectively. Three cases of dengue hemorrhagic fever (DHF) were reported in Guadeloupe, two of which resulted in dengue shock syndrome (DSS) [22]. In Martinique, 47 DHF cases were recorded, involving 17 DSS cases. The epidemic profile differed between the two islands; DENV-2 was mainly reported by Santé Publique France in Guadeloupe (65%), whereas it was DENV-3 in Martinique (92%). Unfortunately, the circulation of DENV serotypes and genotypes is poorly known in the Lesser Antilles [22]. To cope with this lack of knowledge, in this study we (i) characterized DENV serotypes of 3228 samples collected during the 2019–2021 outbreak in Guadeloupe and Martinique, (ii) identified by complete genome sequencing of 32 samples the circulating DENV genotypes, and (iii) performed phylogenetic investigations to better understand the circulation dynamics of DENV during the 2019–2021 outbreak in Guadeloupe and Martinique islands.

## 2. Material and Methods

### 2.1. Ethic Statement

Samples involved in this study were chosen among human serum specimens received as part of routine diagnostic and epidemiological surveillance activities of the laboratory surveillance network (Institut Pasteur de la Guadeloupe (IPG), University Hospitals of Guadeloupe (Centre Hospitalier Universitaire de la Guadeloupe; CHUG), and University Hospitals of Martinique (Centre Hospitalier Universitaire de la Martinique; CHUM)) of the “Comité d’experts des maladies infectieuses et émergentes de Guadeloupe et Martinique”. The study was non-interventional, e.g., with no additional samplings or specific procedures for subjects. Information on the secondary use of diagnostic samples is provided in the patient booklet at the university hospitals of Guadeloupe and Martinique. All samples were rendered anonymous and renumbered prior to preparation of extracted RNA for sequencing. Internal ethics committee of the Pasteur Institute of Guadeloupe has approved this study.

### 2.2. Human Sample Collection and Dengue Virus Screening

During the last dengue epidemic of 2019–2021, 3228 available sera from dengue cases confirmed by the IPG, the CHUG, and the CHUM were used. Venous blood samples (5 mL) from symptomatic patients were collected by several laboratories in the frame of their diagnostic activities and stored at 4 °C during the transportation to the laboratories.

In Guadeloupe, serum samples from CHUG and IPG were first submitted to nucleic acid extraction using the NucleoSpin^®^ 96 Virus kit (Macherey Nagel, Duren, Germany) at IPG, following the manufacturer’s instructions, and then screened for presence of DENV by real-time RT-PCR using primers previously described [25]. If samples were DENV-positive, serotyping was conducted using specific real-time RT-PCR primers [26,27]. The real-time RT-PCRs were performed using Superscript^®^ III Platinum One step Quantitative RT-PCR System with ROX kit (Invitrogen, Carlsbad, CA, USA) and the 7500 Real-Time PCR (Applied Biosystems, Waltham, MA, USA).

In Martinique, nucleic acids were extracted at CHUM from plasma using the NucliSens easyMag automated system (BioMérieux, Craponne, France). Samples were then screened by real-time RT-PCR using the Simplexa^TM^ Dengue Kit (DiaSorin Molecular, Cypress, CA, USA) for virus detection and serotype determination.

### 2.3. Viral Genome Sequencing

Eight overlapping amplicons were produced using the SuperScript^®^ IV One-Step RT-PCR (Thermo Fisher, Waltham, MA, USA) and primers described by Baronti et al. [28]. PCR products were pooled in equimolar proportions. After quantification (Qubit^®^ dsDNA HS Assay Kit and Qubit 4.0 fluorometer (Thermo Fisher, Waltham, MA, USA), amplicons were sonicated (Bioruptor^®^, Diagenode, Liège, Belgium) into 250 pb long fragments. Libraries were built adding to fragmented DNA barcode for sample identification and primers with Ion Plus Fragment Library Kit using AB Library Builder System (Thermo Fisher, Waltham, MA, USA). To equimolarly pool the barcoded samples, a real-time PCR quantification step was performed using Ion Library TaqMan™ Quantitation Kit (Thermo Fisher). Next steps included an emulsion PCR of the pools and loading on 530 chips performed using the automated Ion Chef instrument (Thermo Fisher, Waltham, MA, USA), followed by sequencing using the S5 Ion torrent technology (Thermo Fisher, Waltham, MA, USA), following the manufacturer’s instructions. Consensus sequence was obtained after trimming of reads (reads with quality score < 0.99 and length < 100 pb were removed, and the 30 first and 30 last nucleotides were removed from the reads) and mapping of the reads on a reference (OM909246, MN272404, MN018385, ON908232, MT261978 for inoculum strain and inoculum strain for samples) using CLC genomics workbench software v.21.0.5 (Qiagen, Hilden, Germany). Parameters for reference-based assembly consisted of match score = 1, mismatch cost = 2, length fraction = 0.5, similarity fraction = 0.8, insertion cost = 3, and deletion cost = 3. A de novo contig was also produced to ensure that the consensus sequence was not affected by the reference sequence. 

### 2.4. Recombination Study

The occurrence of molecular recombination was investigated for each alignment of complete CDS using the Recombination Detection Program (RDP) version 4 software [29]. RDP, GENCONV, and MAXCHI methods were used for primary screening, and BOOTSCAN and SISCAN methods were used to check for recombination signals [30,31,32,33]. For optimal recombination detection, the automask procedure was selected. Recombination events with an average *p*-value with RDP lower than E-10 were selected for downstream phylogenetic analyses.

### 2.5. Phylogenetic Analysis

Whole-genome consensus sequences of each serotype were aligned with reference sequences downloaded from GenBank in May 2023, according to their isolation date and locality. They were then aligned using the Clustal W multiple sequence alignment software integrated into BioEdit 7.2.5 (Manchester, UK). The best-fit nucleotide substitution pattern was determined using MEGA 7.0.26 [34] according to the corrected Akaike information criterion (AICc) [35]. Maximum likelihood phylogenetic trees were then constructed with the GTR + G + I model, corresponding to the best-fit model, and 1000 bootstrap replicates were generated [36]. A sequence identity matrix was also generated to determine the proportion of identical residues between the different sequences using BioEdit 7.2.5 [34]. The sequences produced in this study (OR229954-OR229985; Table 1) were compared to 48 reference genomes retrieved from GenBank database for DENV-1, to 57 reference genomes for DENV-2, and to 45 reference genomes to DENV-3 (Appendix A).

## 3. Results

### 3.1. Heterogeneity of the Main Dengue Virus Serotype Circulation Confirmed for Guadeloupe and Martinique during the 2019–2021 Outbreak

Investigations were performed in Guadeloupe and Martinique from August 2019 to February 2021 using the serum samples reported as positive for DENV by IPG, CHUG, and CHUM.

In Guadeloupe (IPG and CHUG), 682 cases were confirmed positive for DENV by real-time RT-PCR, and 522 of these were tested for serotype (Figure 1A). This revealed the predominance of DENV-2 (81.8%, N = 427), followed by DENV-1 (11.3%, N = 59) and DENV-3 (6.9%, N = 36). During the 2019–2021 epidemic, the number of cases increased drastically on two occasions: between October 2019 and March 2020 (N = 148), and between August and December 2020 (N = 297) (Figure 1A). Throughout the epidemic, DENV-1 was recorded at a low level, and DENV-3 appeared in the second half of the epidemic in August 2020, with very low circulation until the beginning of 2021 (Figure 1A).

In Martinique (CHUM), 2546 cases were confirmed positive and identified for DENV by real-time RT-PCR (Figure 1B). This revealed the predominance of DENV-3 (91.4%, N = 2327), followed by DENV-2 (7.5%, N = 192) and DENV-1 (1%, N = 25). During the latest epidemic, the number of cases increased dramatically between July and November 2020 (Figure 1B). Throughout the investigation period, DENV-2 was recorded at a low level, and DENV-1 appeared in the second half of the epidemic in April 2020, with low circulation until the end of 2020 (Figure 1B).

Overall, our results confirmed the circulation of different predominant serotypes in each island (DENV-2 in Guadeloupe and DENV-3 in Martinique) during the 2019–2021 dengue epidemic, although DENV-1 was detected at low levels, and DENV-4 was not detected among the samples tested during this period.

### 3.2. Two Origins of Introduction of DENV-1 Genotype V in Guadeloupe

A DENV-1 phylogenetic investigation was performed on four sequences from Guadeloupe with a genome length of 10,665 bp. Due to the paucity of DENV-1 cases in the two islands, we were unable to obtain isolates for sequencing from Martinique (Table 1 and Appendix A). No recombination event was detected between the Guadeloupe sequences or with the reference sequences by RDP analysis. All the sequences from Guadeloupe belonged to DENV-1 genotype V (Figure 2). However, two different clades were recorded (Clade 1 and Clade 2) and could emphasize two different introduction events in this island. Three sequences collected in July and November 2020 were grouped in Clade 1 (IPG 5, IPG 11 and IPG 12) and shared 99.3% similarity. These sequences also shared 99.1% similarity with samples from Caribbean/Central American origin from Florida (OM909246) and Saint Martin (OP895911), respectively isolated in 2020 and 2021. Clade 2 included one isolate from Guadeloupe (IPG 8; October 2020) clustered with sequences of South American origin from Peru, Venezuela, Colombia, and Ecuador. These sequences shared at least 98.1% identity.

### 3.3. Homogenous Circulation of DENV-2 Cosmopolitan Genotype in Guadeloupe and Martinique Islands

Phylogenetic analysis of DENV-2 complete genome (length of 10,667 bp) was conducted on 12 and seven samples collected in Guadeloupe and Martinique, respectively (Table 1 and Appendix A). As for DENV-1, no recombination events were recorded between these sequences and with the reference ones. Analysis revealed that all the DENV-2 sequences belonged to the cosmopolitan genotype, sharing at least 99.6% identity, which was also the minimum of similarity found between the samples from each island (Figure 3). DENV-2 sequences from this study diverged by 0.9% from a 2018 sample from Reunion Island (MN272404) and clustered also (at least 98.9% similarity) with samples from the Seychelles (MN272405), India (MK858107, MK858111), and Singapore (MW512468) isolated between 2016 and 2017 (Figure 3).

### 3.4. Two Different DENV-3 Genotype III Introductions in Guadeloupe and Martinique

The complete genome of a panel of nine DENV-3 sequences (two from Guadeloupe and seven from Martinique), with a length of 10,649 bp, was obtained, and a multiple alignment was performed with GenBank sequences (Table 1 and Appendix A). No recombination events were found between the sequences from Guadeloupe and Martinique, or with those from GenBank, used to construct this phylogenetic tree. All the DENV-3 sequences of the study belonged to genotype III (Figure 4). However, they grouped into two different clades (Clade 1 and Clade 2), with between 2.9% and 3.1% divergence, and grouped under a node with a bootstrap of 100. The CHUM 11, CHUM 21, and CHUM 22 sequences belonged to Clade 1 (99.7% of similarity) and grouped with sequences originating from Africa (Burkina Faso MT261978 and Ethiopia ON890788; 98.8% similarity). Six strains from Martinique and Guadeloupe (CHUM 13, CHUM 16, CHUM 17, CHUM 24, IPG 16, IPG 17) grouped in Clade 2 (99.5% identity). This clade included sequences from Asia (China MN018385, ON890789, Maldives ON890789, and India MK858153, ON123658) collected between 2016 and 2019 (98.6% similarity). The presence of two different clades among DENV-3 sequences might indicate two different introduction events in Martinique.

## 4. Discussion

Over the past two decades, the surveillance systems across the FTA have made continuous improvements and demonstrated an evolution of the epidemiology to a hyperendemic state characterized by an increase in the co-circulation of the four DENV serotypes, with large outbreaks associated with a shift in a predominating serotype [19]. The two islands, Guadeloupe and Martinique, have similar socio-economical, geographical, and environmental characteristics, with comparable availability of French healthcare systems, and a similar population size of 400,000 inhabitants in Guadeloupe and 380,000 in Martinique. They also share similar trends of dengue epidemiology and evolution, including the timing of epidemics and circulating serotypes [19,37]. However, differences in dynamics observed since the 2013–2014 epidemic are more enigmatic given that the distance separating the two islands is only 189 km and that daily air and sea traffic is significant.

The last major epidemic, therefore, began in 2019, with only two months between the alert in Guadeloupe and Martinique and the end declared in 2021. The total duration of the epidemic, 73 weeks for Guadeloupe and 67 weeks for Martinique, made them the longest ever recorded since the implementation of the dengue surveillance in these territories. Furthermore, differences in circulation dynamics were observed, with two distinct epidemic peaks in Guadeloupe but only one in Martinique. The context of COVID-19 pandemic may have played a role in these historical durations associated with persistent but lower daily incidence rates. The general rule of simultaneous occurrence of dengue epidemics between Guadeloupe and Martinique was found in 2019; however, the serotype distributions were different with large predominance DENV-2 in Guadeloupe and DENV-3 in Martinique. Furthermore, during the same period, DENV-1 was the predominant serotype detected in two other French Caribbean islands (Saint-Martin and Saint-Barthelemy), during the dengue outbreak [22]. Therefore, we can state that these islands have experienced partitioned epidemics.

At a genetic level, DENV-1 samples collected in Guadeloupe during the 2019–2021 were identified as genotype V, probably introduced from Southeast Asia [38] or India [39]. On the American continent, the Caribbean could be assimilated as a hub for the circulation of DENV-1 genotype V since 1980 and its circulation in South America [40]. Interestingly, in our study, two different clades were recorded (Clade 1 and Clade 2). The presence of two clades during this 2019–2021 outbreak may be linked to two independent events as previously described in Asia [41,42,43]. Clade 1 grouped with samples with Caribbean/Central American origins from Saint Martin and Florida collected in 2020 and 2021 and emphasize an active circulation of the clade in this region. Clade 2, corresponding to an isolate from Guadeloupe collected in 2020 (IPG 8), grouped with strains from South America. This strain is linked to other sequences which circulated in Peru, Venezuela, and Colombia between 2015 and 2021, sharing 98.1% identity. DENV-1 genotype V is known as the predominant DENV-1 genotype, which has circulated in South America for over 50 years [40]. During the latest epidemics in Colombia and Venezuela, which occurred in 2015–2016 and 2018–2019, the DENV-1 strains isolated also belonged to genotype V [44]. Although the DENV-1 serotype was recorded at a low level in both Guadeloupe and Martinique islands, all these phylogenetic data revealed the active circulation of this serotype and especially genotype V in Americas.

DENV-2 was the predominant serotype circulating in Guadeloupe, whereas it was detected at a low level in Martinique. In both islands, genetic data revealed the presence of the cosmopolitan genotype only. All the strains from the two territories were, therefore, genetically homogeneous and belonged to the same clade. The 19 strains sequenced showed a high homology with sequences from the Indian Ocean (Reunion Island and the Seychelles) and India, which circulated between 2016 and 2018 (98.9%). Unfortunately, as there were no recent sequences published on GenBank, we could not determine the importation origin of these strains into the two French territories. This may be an indirect consequence of the COVID-19 pandemic, as, since the end of 2019, all sequencing efforts appeared to be mainly focused on this virus [45].

DENV-3 results demonstrated the homogeneous presence of DENV-3 genotype III in Guadeloupe and Martinique islands. Two clades within genotype III were detected among the samples sequenced (Clade I and Clade 2), and these clades were both detected in samples collected in 2020. Clade I, which grouped samples from Guadeloupe and Martinique, appeared to link with samples from Africa (Ethiopia, Burkina Faso, and Gabon), collected between 2016 and 2019. This genotype is known to be widespread in Africa since 2010, especially in countries of East Africa such as Sudan, Kenya, or Tanzania [46,47,48,49,50]. Clade II, corresponding to sequences collected in Martinique, clustered with sequences from China, the Maldives, and India collected between 2016 and 2019. DENV-3 genotype III is also known to be endemic to South and Southeast Asia [51], thus being responsible for numerous epidemics such as that which occurred in Lao PDR in 2012 [52], in China in 2019 [53], and in New Delhi, India, in 2016 [54]. The presence of two clades in these islands emphasize the multiple introductions of DENV-3 in FTA during the 2019–2021 outbreak.

Our results demonstrated multiple introductions of DENV in both Guadeloupe and Martinique islands. As other Caribbean islands, the FTA are highly touristic, with tourist numbers rising steadily in Martinique and Guadeloupe since 2006 [55]. This phenomenon could promote the risk of arbovirus introduction in these islands as previously described. Indeed, in 2013, a chikungunya outbreak was reported in the French Caribbean, initially in Saint-Martin, and then rapidly in Martinique and Guadeloupe, before spreading throughout the Americas reflecting the potential role of small touristic archipelago as a hub between continental territories for arboviral infections of global public health significance [56,57]. However, the 2019–2021 dengue outbreaks in FTA were correlated with the emergence and spread of the COVID-19 pandemic [58]. Indeed, the first detections of SARS-CoV-2 were reported in March 2020 in Guadeloupe and Martinique [58]. The inter-island travel restrictions associated with the COVID-19 waves may have reduced the number of founder opportunities and impaired DENV-3 development in Guadeloupe. Indeed, the impact of this pandemic, especially the lockdowns, on dengue outbreak is still misunderstood [59]. The global burden of dengue in this region, in the context of the COVID-19 pandemic, must be investigated to determine if the circulation of SARS-CoV-2 had an impact on DENV circulation, as well as on DENV diagnosis and reporting [60,61]. Differences in the herd immunity of populations since the 2010s may also play a fundamental role in dengue dynamics in both territories. Despite the high proportion of the population of French Caribbean islands found to be positive for DENV antibodies [19], the impact of pre-existing, cross-reactive, and waning memory immunity on subsequent dengue infections and outbreak dynamics remains poorly understood.

Considering epidemiologic and genetic data, the heterogenous circulation of DENV serotypes in FTA could be influenced by the presence of *Ae. Aegypti* mosquito in both islands and, perhaps, to a differential ability to transmit each DENV serotype. Indeed, DENV transmission by *Ae. aegypti* is associated with genotype–genotype interactions [62]. Previous studies demonstrated that Caribbean *Ae. aegypti* populations could be genetically different [63]. Differences in genetic backgrounds could influence DENV transmission and be the origin of heterogenous DENV serotypes circulation in Guadeloupe and in Martinique [62,64]. Furthermore, the co-circulation of DENV serotypes may promote co-infection in the vector and may induce viral competition in the *Ae. aegypti* midgut as previously described for DENV-1 and DENV-4 and mosquitoes from French Guiana [65].

In conclusion, our study has described for the first time the genetic diversity of DENV serotypes in Guadeloupe and Martinique, with a focus on 2019–2021 epidemic. This is a first step for the genetic and phylogenetic characterization of DENV in this region, where dengue has been a major public health problem for more than 50 years. Prospective genomic surveillance may be useful to increase knowledge on dengue circulation dynamics and preparedness. It is, therefore, crucial to combine multidisciplinary approaches to assess the influence of various factors, such as those associated with the host, the virus, or virus/vector interactions on dengue transmission.

## Figures and Tables

**Figure 1 pathogens-12-01182-f001:**
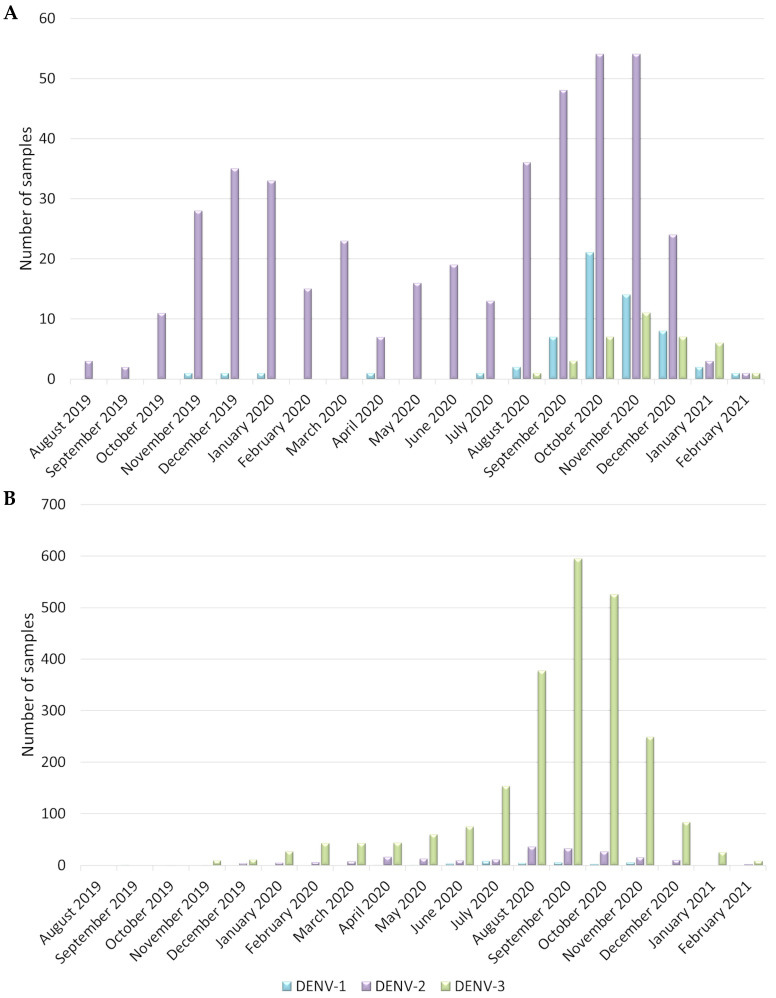
Dynamics of DENV circulation in (**A**) Guadeloupe (N = 682) and (**B**) Martinique (N = 2546) during 2019–2021 epidemic.

**Figure 2 pathogens-12-01182-f002:**
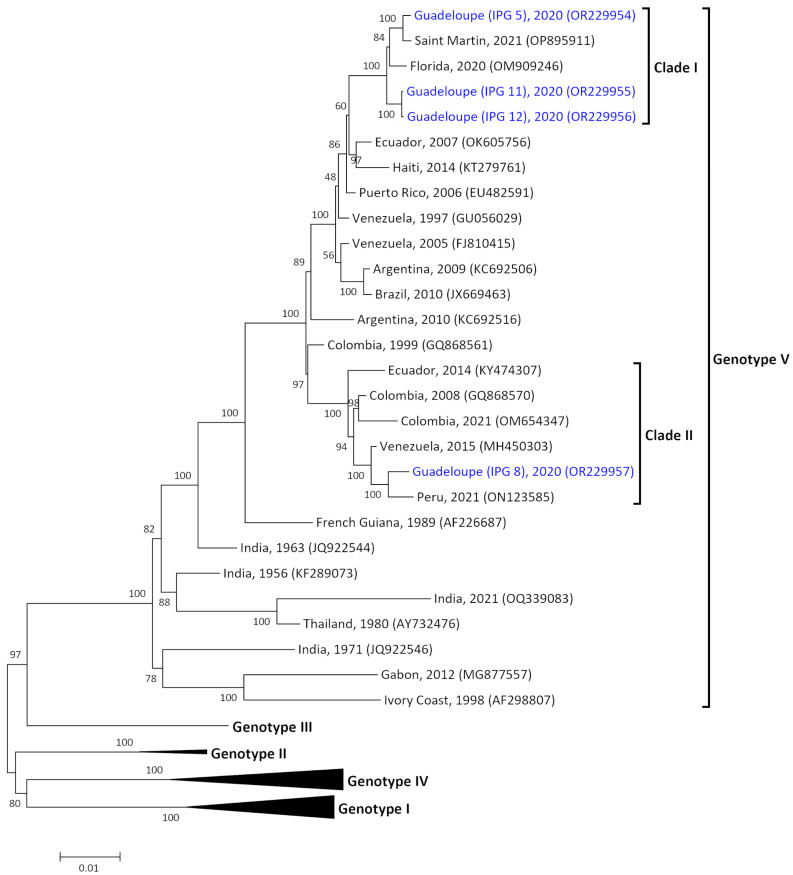
Maximum likelihood phylogenetic tree of DENV-1 from French Territories of America. Tree was built with complete genomes (length 10,665 bp). Bootstrap values are shown on branch nodes. Scale bar indicates the nucleotide substitution per site. The Guadeloupe strains sequenced in this study are indicated in blue.

**Figure 3 pathogens-12-01182-f003:**
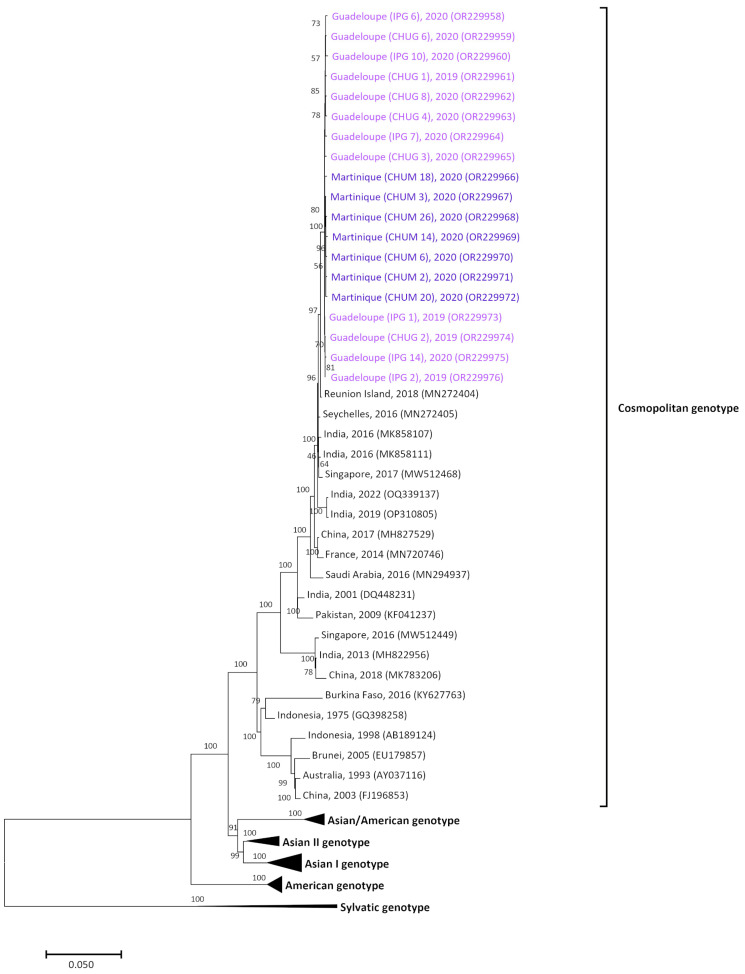
Maximum likelihood phylogenetic tree of DENV-2 from French Territories of America. Tree was built with complete genomes (length 10,667 bp). Bootstrap values are shown on branch nodes. Scale bar indicates the nucleotide substitution per site. The Guadeloupe strains sequenced in this study are indicated in dark purple, and the Martinique strains in light purple.

**Figure 4 pathogens-12-01182-f004:**
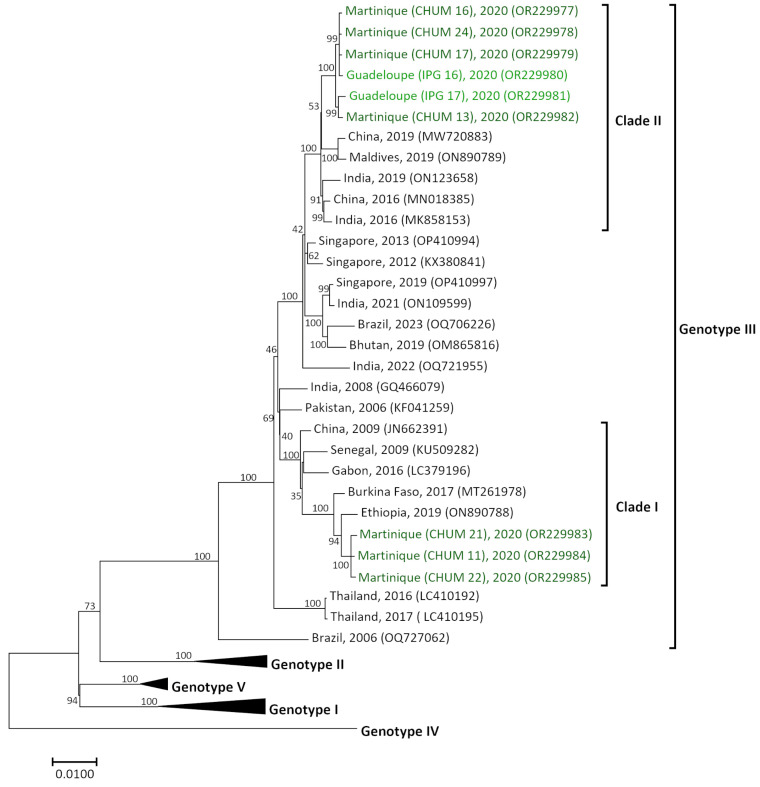
Maximum likelihood phylogenetic tree of DENV-3 from Guadeloupe and Martinique. The tree was built with complete genomes (length 10,649 bp). Bootstrap values are shown on branch nodes. The scale bar indicates the nucleotide substitution per site. The Guadeloupe strains sequenced in this study are indicated in dark green, and the Martinique strains are in light green.

**Table 1 pathogens-12-01182-t001:** Guadeloupe and Martinique dengue sequences obtained in this study.

Sample Identification	Number of Nucleotides	Origin	Year of Collection	Isolation Source	Serotype	Genbank Number
**Guadeloupe (IPG5), 2020**	10,665	Guadeloupe	July 2020	Plasma	DENV-1	OR229954
**Guadeloupe (IPG8), 2020**	10,665	Guadeloupe	October 2020	Plasma	DENV-1	OR229957
**Guadeloupe (IPG11), 2020**	10,665	Guadeloupe	November 2020	Plasma	DENV-1	OR229955
**Guadeloupe (IPG12), 2020**	10,665	Guadeloupe	November 2020	Plasma	DENV-1	OR229956
**Guadeloupe (CHUG1), 2020**	10,667	Guadeloupe	October 2019	Plasma	DENV-2	OR229961
**Guadeloupe (CHUG2), 2019**	10,667	Guadeloupe	December 2019	Plasma	DENV-2	OR229974
**Guadeloupe (CHUG3), 2020**	10,667	Guadeloupe	January 2020	Plasma	DENV-2	OR229965
**Guadeloupe (CHUG4), 2020**	10,667	Guadeloupe	July 2020	Plasma	DENV-2	OR229963
**Guadeloupe (CHUG6), 2020**	10,667	Guadeloupe	September 2020	Plasma	DENV-2	OR229959
**Guadeloupe (CHUG8), 2020**	10,667	Guadeloupe	October 2020	Plasma	DENV-2	OR229962
**Guadeloupe (IPG1), 2020**	10,667	Guadeloupe	September 2019	Plasma	DENV-2	OR229973
**Guadeloupe (IPG2), 2020**	10,667	Guadeloupe	November 2019	Plasma	DENV-2	OR229976
**Guadeloupe (IPG6), 2020**	10,667	Guadeloupe	July 2020	Plasma	DENV-2	OR229958
**Guadeloupe (IPG7), 2020**	10,667	Guadeloupe	August 2020	Plasma	DENV-2	OR229964
**Guadeloupe (IPG10), 2020**	10,667	Guadeloupe	October 2020	Plasma	DENV-2	OR229960
**Guadeloupe (IPG14), 2020**	10,667	Guadeloupe	November 2020	Plasma	DENV-2	OR229975
**Martinique (CHUM2), 2020**	10,667	Martinique	June 2020	Plasma	DENV-2	OR229971
**Martinique (CHUM3), 2020**	10,667	Martinique	June 2020	Plasma	DENV-2	OR229967
**Martinique (CHUM6), 2020**	10,667	Martinique	June 2020	Plasma	DENV-2	OR229970
**Martinique (CHUM14), 2020**	10,667	Martinique	July 2020	Plasma	DENV-2	OR229969
**Martinique (CHUM18), 2020**	10,667	Martinique	August 2020	Plasma	DENV-2	OR229966
**Martinique (CHUM20), 2020**	10,667	Martinique	August 2020	Plasma	DENV-2	OR229972
**Martinique (CHUM26), 2020**	10,667	Martinique	September 2020	Plasma	DENV-2	OR229968
**Guadeloupe (IPG16), 2020**	10,649	Guadeloupe	December 2020	Plasma	DENV-3	OR229980
**Guadeloupe (IPG17), 2020**	10,649	Guadeloupe	October 2020	Plasma	DENV-3	OR229981
**Martinique (CHUM11), 2020**	10,649	Martinique	July 2020	Plasma	DENV-3	OR229984
**Martinique (CHUM13), 2020**	10,649	Martinique	July 2020	Plasma	DENV-3	OR229982
**Martinique (CHUM16), 2020**	10,649	Martinique	August 2020	Plasma	DENV-3	OR229977
**Martinique (CHUM17), 2020**	10,649	Martinique	August 2020	Plasma	DENV-3	OR229979
**Martinique (CHUM21), 2020**	10,649	Martinique	September 2020	Plasma	DENV-3	OR229983
**Martinique (CHUM22), 2020**	10,649	Martinique	September 2020	Plasma	DENV-3	OR229985
**Martinique (CHUM24), 2020**	10,649	Martinique	September 2020	Plasma	DENV-3	OR229978

IPG: Institut Pasteur de la Guadeloupe; CHUG: Centre Hospitalier Universitaire de la Guadeloupe, University Hospitals of Guadeloupe; CHUM: Centre Hospitalier Universitaire de la Martinique, University Hospitals of Martinique.

## Data Availability

Not applicable.

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
