# Peer review of "Phylogenetic Investigations of Dengue 2019–2021 Outbreak in Guadeloupe and Martinique Caribbean Islands"

_pathogens, 2023, doi:10.3390/pathogens12091182_

Round 1

Reviewer 1 Report

I am attaching a PDF file (20230901_pathogens2531474_review.pdf). In there, the authors will see highlighted text. All highlighted text comes with comments. The authors should find and read each comment carefully. I indicate which are suggestions and which are not. The others point to components of the manuscript that must be improved before publication.

The manuscript needs a description of the methodology and incorrectly presents results from phylogenetic experiments. After reviewing my comments on the highlighted text, authors must:

1) The methodology for sequence alignment must be presented in complete detail. The final alignment and results trees should be presented in the supplementary material as text files.

2) The methodology for sequence alignment must be presented in complete detail. Tree search parameters must be included. Bootstrap calculation parameters must be included. There should be a justification for the choice of phylogenetic criterion.

3) Please show all bootstrap values and remember that bootstrap is a measure of clade frequency not correlated to clade support in maximum likelihood. The 70% threshold on bootstrap values is meaningless and historically linked to the misinterpretation of a paper by  Hillis & Bull (1993).

Author Response

Reviewer 1

Comments and Suggestions for Authors

I am attaching a PDF file (20230901_pathogens2531474_review.pdf). In there, the authors will see highlighted text. All highlighted text comes with comments. The authors should find and read each comment carefully. I indicate which are suggestions and which are not. The others point to components of the manuscript that must be improved before publication.

We would like to thank the reviewer for his interesting comments. We modified the manuscript according to the suggestions and the comments provided by the reviewers. We carefully checked the pdf document updated by the Reviewer. Although some modifications were directly added to the manuscript, we also answered to the reviewer’ comments below:

Line 70: We reviewed all the manuscript and we added a space before all the references.

Line 143: We agreed with the reviewer’ comment and we added the information concerning the whole genome analysis.

“Whole genome consensus sequences of each serotype were aligned with reference sequences downloaded from GenBank according to their isolation date and locality.”

Line 144: As recommended by the reviewer we clarified this section.

“The best-fit nucleotide substitution pattern was determined using MEGA 7.0.26 [35] according to the corrected Akaike information criterion (AICc) [36]. Maximum likelihood phylogenetic trees were then constructed with the GTR+G+I model, corresponding to the best-fit model, and 1000 bootstrap replicates were generated [37].”

Line 146: The phylogenetic trees were generated using the whole genome consensus sequences for each serotype as previously described in publications (1-4). We modified the manuscript according to this comment:

“The sequences produced in this study (OR229954-OR229985) (Table 1) were compared to 48 reference genomes retrieved from GenBank database for DENV-1, to 57 for DENV-2, and to 45 references to DENV-3 (Table S1).”

  1. Carrillo-Hernandez, M.Y.; Ruiz-Saenz, J.; Jaimes-Villamizar, L.; Robledo-Restrepo, S.M.; Martinez-Gutierrez, M. Phylogenetic and evolutionary analysis of dengue virus serotypes circulating at the Colombian–Venezuelan border during 2015–2016 and 2018–2019. PLOS ONE 2021, 16, e0252379, doi:10.1371/journal.pone.0252379.
  2. Dussart, P.; Lavergne, A.; Lagathu, G.; Lacoste, V.; Martial, J.; Morvan, J.; Cesaire, R. Reemergence of Dengue Virus Type 4, French Antilles and French Guiana, 2004–2005. Emerg Infect Dis 2006, 12, 1748–1751, doi:10.3201/eid1211.060339.
  3. Eldigail, M.H.; Abubaker, H.A.; Khalid, F.A.; Abdallah, T.M.; Musa, H.H.; Ahmed, M.E.; Adam, G.K.; Elbashir, M.I.; Aradaib, I.E. Association of Genotype III of Dengue Virus Serotype 3 with Disease Outbreak in Eastern Sudan, 2019. Virol J 2020, 17, 118, doi:10.1186/s12985-020-01389-9.
  4. Cao-Lormeau, V.-M.; Roche, C.; Aubry, M.; Teissier, A.; Lastere, S.; Daudens, E.; Mallet, H.-P.; Musso, D.; Aaskov, J. Recent Emergence of Dengue Virus Serotype 4 in French Polynesia Results from Multiple Introductions from Other South Pacific Islands. PLoS One 2011, 6, e29555, doi:10.1371/journal.pone.0029555.

Line 175: As recommended by the reviewer, we increased the font size and the bars width on figure 1. We kept the colors in the graph to be homogenous with the color-code used for DENV serotypes throughout all the manuscript figures, including trees.

Ligne 177: Following reviewer’ recommendation, the second part of this paragraph has been moved to the discussion section.

“Overall, our results confirmed the circulation of different predominant serotypes in each island (DENV-2 in Guadeloupe and DENV-3 in Martinique) during the 2019-2021 dengue epidemic. Although DENV-1 was detected at low levels, DENV-4 was not detected among the samples tested during this period.”

Phylogenetic trees and legends: According to the reviewer’ recommendation we modified the legend and the phylogenetic trees.

Line 222, 229: We carefully checked the percentage of similarity between the two clades, and found a divergence of around 3%, which supports our hypothesis. Furthermore, each clade grouped under a node with bootstrap 100. However, we have modulated our remarks thanks to the reviewer's comments.

“However, they grouped into two different clades (Clade 1 and Clade 2), with between 2.9% and 3.1% of divergence, and grouped under a node with a bootstrap 100. The sequences CHUM 11, CHUM 21 and CHUM 22 belonged to Clade 1 (99.7% of similarity) and grouped with sequences originating from Africa (Burkina Faso MT261978, and Ethiopia ON890788; 98.8% of similarity). Six strains from Martinique and Guadeloupe (CHUM 13, CHUM 16, CHUM 17, CHUM 24, IPG 16, IPG 17) grouped in Clade 2 (99.5% of identity). This clade included sequences from Asia (China MN018385, ON890789, Maldives ON890789 and India MK858153, ON123658) collected between 2016 and 2019 (98.6% of similarity). The presence of two different clades among DENV-3 sequences might indicate two different introduction events in Martinique.”

Line 237 to 263: We shortened the first two paragraphs as requested and modified the following sentences (lines 246-250, line 255):

“However, differences in dynamics observed since the 2013-2014 epidemic are more enigmatic given that the distance separating the two islands is only 189 km and that daily air and sea traffic is significant.”

“Furthermore, differences in circulation dynamics were observed, with two distinct epidemic peaks observed in Guadeloupe but only one in Martinique.”

The manuscript needs a description of the methodology and incorrectly presents results from phylogenetic experiments. After reviewing my comments on the highlighted text, authors must:

1) The methodology for sequence alignment must be presented in complete detail. The final alignment and results trees should be presented in the supplementary material as text files.

According to the reviewer comment, we updated the methodology part in the manuscript:

“Whole genome consensus sequences of each serotype were aligned with reference sequences downloaded from GenBank in May 2023, according to their isolation date and locality. They were then aligned using the Clustal W multiple sequence alignment software integrated into BioEdit 7.2.5 (Manchester, UK). The best-fit nucleotide substitution pattern was determined using MEGA 7.0.26 [35] according to the corrected Akaike information criterion (AICc) [36]. Maximum likelihood phylogenetic trees were then constructed with the GTR+G+I model, corresponding to the best-fit model, and 1000 bootstrap replicates were generated [37]. Sequence identity matrix was also generated to determine the proportion of identical residues between the different sequences using BioEdit 7.2.5 [35]. The sequences produced in this study (OR229954-OR229985; Table 1) were compared to 48 reference genomes retrieved from GenBank database for DENV-1, to 57 for DENV-2, and to 45 references to DENV-3 (Table S1).”

2) The methodology for sequence alignment must be presented in complete detail. Tree search parameters must be included. Bootstrap calculation parameters must be included. There should be a justification for the choice of phylogenetic criterion.

According to this comment we modified this section of the manuscript:

“Whole genome consensus sequences of each serotype were aligned with reference sequences downloaded from GenBank in May 2023, according to their isolation date and locality. They were then aligned using the Clustal W multiple sequence alignment software integrated into BioEdit 7.2.5 (Manchester, UK). The best-fit nucleotide substitution pattern was determined using MEGA 7.0.26 [35] according to the corrected Akaike information criterion (AICc) [36]. Maximum likelihood phylogenetic trees were then constructed with the GTR+G+I model, corresponding to the best-fit model, and 1000 bootstrap replicates were generated [37]. Sequence identity matrix was also generated to determine the proportion of identical residues between the different sequences using BioEdit 7.2.5 [35]. The sequences produced in this study (OR229954-OR229985; Table 1) were compared to 48 reference genomes retrieved from GenBank database for DENV-1, to 57 for DENV-2, and to 45 references to DENV-3 (Table S1).”

3) Please show all bootstrap values and remember that bootstrap is a measure of clade frequency not correlated to clade support in maximum likelihood. The 70% threshold on bootstrap values is meaningless and historically linked to the misinterpretation of a paper by Hillis & Bull (1993).

According to the reviewer comment, we updated our phylogenetic trees by the addition of all the bootstrap values. Please note we only discuss results relying in bootstraps superior to 90%.

Reviewer 2 Report

In this study, the authors obtained complete genome sequences of Dengue virus (DENV) from serum samples of infected patients and carried out phylogenetic analyses to characterize the circulating DENV strains during the outbreaks in Martinique and Guadeloupe in 2019-2021. The study is of high interest to the scientific community as DENV is a potentially life-threatening disease, and prophylactic measures or specific therapeutic options against DENV infection are still missing. In addition, no genetic sequences of circulating DENV were available for this geographic area so far.

The manuscript is well written and easy to read. I just have a few comments that will hopefully be considered to improve the manuscript.

Major comment:

My major concern is the methodology used for the phylogenetic inference. Recombination events were widely described in literature for different serotypes of DENV (Holmes EC et al., 1999; Worobey M et al., 1999; Tolou et al., 2001; Uzcategui et al., 2001; Chen et al., 2008; Perez-Ramirez et al., 2009; Weaver&Vasilakis, 2009; Wu W et al., 2011; Sun B et al., 2020), which play a crucial role in driving DENV evolution. In presence of recombination, different parts of the sequence have different phylogenetic history. Therefore, in case breakpoints are present within the genomic sequences, a phylogenetic reconstruction based on the whole genome could produce artefacts.

I highly recommend to carry out a recombination analysis on the ClustalW alignments (within the same serotype but possibly also between different serotypes). If recombination events are detected, then the whole genome should be partitioned and the phylogenetic analysis should be done separately for each partition. Only then solid conclusions about the genetic diversity and multiple introductions of DENV in Martinique and Guadeloupe can de drawn.

Minor comments:

Line 58: “they are” should be replaced with “there are”

Lines 59-60: this sentence should be rephrased including the term “cross-protection”

Lines 61-62: the authors should add one sentence in the introduction to describe the different DENV genotypes, as they are cited later in the text (i.e. cosmopolitan, sylvatic). This would make it easier for the reader to follow.

Lines 103-110: please include the number of tested sera in Guadeloupe and Martinique in the material and methods section

Lines 111, 117, 119, 125: please homogenize the product details throughout the material and methods section, sometimes the country is present and sometimes not

Line 141: please specify in material and methods that DENV complete genome sequences were submitted to NCBI

Lines 143-145: this sentence is not clear; was the alignment was produced only with consensus sequences obtained in your study or also publicly available sequences downloaded from NCBI? Please clarify this point

Line 143: this section should include more details about the dataset used for the phylogenetic analysis, as DENV phylogeny is the main focus of the manuscript. In total, how many sequences were aligned, including sequences generated in this study and sequenced downloaded from NCBI?

Line 145: please specify here if the alignment was done on the whole genome sequences

Line 146: Please specify the statistical method used for the phylogenetic analysis, I guess the 1000 replicates refer to Bootstrap method

Line 184: Was the virus also isolated in this study? If not, “isolates” should be replaced with “sequences” throughout the manuscript

Line 189: the authors report the percentages of genetic similarity between obtained DENV sequences, however this analysis was not mentioned in the material and methods section. Please include how the genetic similarity in nucleotides was estimated in M&M.

Lines 229-230: this sentence should be moved in the discussion section

Minor English editing required

Author Response

Reviewer 2

Comments and Suggestions for Authors

In this study, the authors obtained complete genome sequences of Dengue virus (DENV) from serum samples of infected patients and carried out phylogenetic analyses to characterize the circulating DENV strains during the outbreaks in Martinique and Guadeloupe in 2019-2021. The study is of high interest to the scientific community as DENV is a potentially life-threatening disease, and prophylactic measures or specific therapeutic options against DENV infection are still missing. In addition, no genetic sequences of circulating DENV were available for this geographic area so far.

The manuscript is well written and easy to read. I just have a few comments that will hopefully be considered to improve the manuscript.

Many thanks to the reviewer for this very interesting feedback. We have taken account to the comments for improving the manuscript.

Major comment:

My major concern is the methodology used for the phylogenetic inference. Recombination events were widely described in literature for different serotypes of DENV (Holmes EC et al., 1999; Worobey M et al., 1999; Tolou et al., 2001; Uzcategui et al., 2001; Chen et al., 2008; Perez-Ramirez et al., 2009; Weaver&Vasilakis, 2009; Wu W et al., 2011; Sun B et al., 2020), which play a crucial role in driving DENV evolution. In presence of recombination, different parts of the sequence have different phylogenetic history. Therefore, in case breakpoints are present within the genomic sequences, a phylogenetic reconstruction based on the whole genome could produce artefacts.

I highly recommend to carry out a recombination analysis on the ClustalW alignments (within the same serotype but possibly also between different serotypes). If recombination events are detected, then the whole genome should be partitioned and the phylogenetic analysis should be done separately for each partition. Only then solid conclusions about the genetic diversity and multiple introductions of DENV in Martinique and Guadeloupe can de drawn.

Thank you for this valuable comment. According to your suggestion, we reviewed the methodology and used the RDP software to see if the sequences presented recombination events and no recombination event was detected among the sequences obtained in this study or in the reference ones. We added a paragraph on the recombination study in the Material and Methods section and, for our sequences, in the Results section.

“Occurrence of molecular recombination was investigated for each alignment of complete CDS using the Recombination Detection Program (RDP) version 4 software [30]. RDP, GENCONV and MAXCHI methods were used for primary screening and BOOTSCAN and SISCAN methods were used to check for recombination signals [31-34]. For optimal recombination detection, the automask procedure was selected. Recombination events with an average p-value with RDP lower than E-10 were selected for downstream phylogenetic analyses.”

Minor comments:

Line 58: “they are” should be replaced with “there are”

We made the modification.

“There are four antigenically and genetically distinct serotypes (DENV-1, DENV-2, DENV-3 and DENV-4) which emerged from sylvatic cycles in Asia”.

Lines 59-60: this sentence should be rephrased including the term “cross-protection”

We understand the reviewer’ comment and we modified this sentence accordingly:

“An infection with one DENV serotype confers a specific immunity against this serotype but no cross-protection against the other serotypes.”

Lines 61-62: the authors should add one sentence in the introduction to describe the different DENV genotypes, as they are cited later in the text (i.e. cosmopolitan, sylvatic). This would make it easier for the reader to follow.

For clarity, we added this information in the text:

“For DENV-1 and DENV-3 serotypes, five genotypes numbered from Genotype I to V were reported. For DENV-2, six genotypes were recorded and named Asian I, Asian II, Cosmopolitan, Asian/American, American, and Sylvatic). Finally, for DENV-4, four genotypes were described (Genotype I to IV).”

Lines 103-110: please include the number of tested sera in Guadeloupe and Martinique in the material and methods section.

It has been done

“During the last dengue epidemic of 2019-2021, 3,228 available sera from dengue cases confirmed by the IPG, the CHUG and the CHUM were used.”

Lines 111, 117, 119, 125: please homogenize the product details throughout the material and methods section, sometimes the country is present and sometimes not.

We homogenized the product details throughout the manuscript.

Line 141: please specify in material and methods that DENV complete genome sequences were submitted to NCBI

It has been done.

“The sequences produced in this study (OR229954-OR229985) (Table 1) were compared to 48 reference genomes retrieved from GenBank database for DENV-1, to 57 for DENV-2, and to 45 references to DENV-3 (Table S1).”

Lines 143-145: this sentence is not clear; was the alignment was produced only with consensus sequences obtained in your study or also publicly available sequences downloaded from NCBI? Please clarify this point

We added information in the Material and Method part for clarification:

“Whole genome consensus sequences of each serotype were aligned with reference sequences downloaded from GenBank in May 2023 according to their isolation date and locality. They were then aligned using the Clustal W multiple sequence alignment software integrated into BioEdit 7.2.5 (Manchester, UK). The best-fit nucleotide substitution pattern was determined using MEGA 7.0.26 [35] according to the corrected Akaike information criterion (AICc) [36]. Maximum likelihood phylogenetic trees were then constructed with the GTR+G+I model, corresponding to the best-fit model, and 1000 bootstrap replicates were generated [37]. Sequence identity matrix was also generated to determine the proportion of identical residues between the different sequences using BioEdit 7.2.5 [35]. The sequences produced in this study (OR229954-OR229985; Table 1) were compared to 48 reference genomes retrieved from GenBank database for DENV-1, to 57 for DENV-2, and to 45 references to DENV-3 (Table S1).”

Line 143: this section should include more details about the dataset used for the phylogenetic analysis, as DENV phylogeny is the main focus of the manuscript. In total, how many sequences were aligned, including sequences generated in this study and sequenced downloaded from NCBI?

We updated this part of the manuscript according to the reviewer comment:

“Whole genome consensus sequences of each serotype were aligned with reference sequences downloaded from GenBank in May 2023, according to their isolation date and locality. They were then aligned using the Clustal W multiple sequence alignment software integrated into BioEdit 7.2.5 (Manchester, UK). The best-fit nucleotide substitution pattern was determined using MEGA 7.0.26 [35] according to the corrected Akaike information criterion (AICc) [36]. Maximum likelihood phylogenetic trees were then constructed with the GTR+G+I model, corresponding to the best-fit model, and 1000 bootstrap replicates were generated [37]. Sequence identity matrix was also generated to determine the proportion of identical residues between the different sequences using BioEdit 7.2.5 [35]. The sequences produced in this study (OR229954-OR229985; Table 1) were compared to 48 reference genomes retrieved from GenBank database for DENV-1, to 57 for DENV-2, and to 45 references to DENV-3 (Table S1).”

Line 145: please specify here if the alignment was done on the whole genome sequences.

We added this information:

“Whole genome consensus sequences of each serotype were aligned with reference sequences downloaded from GenBank in May 2023, according to their isolation date and locality.”

Line 146: Please specify the statistical method used for the phylogenetic analysis, I guess the 1000 replicates refer to Bootstrap method.

We modified the phylogenetic analysis section part in the manuscript according to the reviewers’ comments:

“Maximum likelihood phylogenetic trees were then constructed with the GTR+G+I model, corresponding to the best-fit model, and 1000 bootstrap replicates were generated [37].”

Line 184: Was the virus also isolated in this study? If not, “isolates” should be replaced with “sequences” throughout the manuscript

We replaced “isolates” by “sequences” in the manuscript.

Line 189: the authors report the percentages of genetic similarity between obtained DENV sequences, however this analysis was not mentioned in the material and methods section. Please include how the genetic similarity in nucleotides was estimated in M&M.

We added this information in the M&M section:

“Sequence identity matrix was also generated to determine the proportion of identical residues between the different sequences using BioEdit 7.2.5 [35].”

Lines 229-230: this sentence should be moved in the discussion section.

We understand reviewer’ comment, but we though this sentence could be kept in the result part because it is the direct result of the presence of two clades for this serotype. We have therefore reworded the paragraph as follows, in order to modulate our comment.

“However, they grouped into two different clades (Clade 1 and Clade 2), with between 2.9% and 3.1% of divergence, and grouped under a node with a bootstrap 100. The sequences CHUM 11, CHUM 21 and CHUM 22 belonged to Clade 1 (99.7% of similarity) and grouped with sequences originating from Africa (Burkina Faso MT261978, and Ethiopia ON890788; 98.8% of similarity). Six strains from Martinique and Guadeloupe (CHUM 13, CHUM 16, CHUM 17, CHUM 24, IPG 16, IPG 17) grouped in Clade 2 (99.5% of identity). This clade included sequences from Asia (China MN018385, ON890789, Maldives ON890789 and India MK858153, ON123658) collected between 2016 and 2019 (98.6% of similarity). The presence of two different clades among DENV-3 sequences might indicate two different introduction events in Martinique.”

Round 2

Reviewer 1 Report

I commend the authors for improving the manuscript and responding to all of my original comments. However, some significant issues remain. Specifically, phylogenetic methodology is more clearly described, but some details are lacking. For example:

1) Multiple alignment parameters are not detailed.
2) Also, there is no indication in the paper about the outgroup for the phylogenetic analyses and how the root was selected. Outgroups are required for phylogenetic analysis, and rooting is needed to determine clades and character transformation's direction.
3) Finally, on the images for the trees, the caption does not wholly describe the image. For example, the values on the branches are not named in the caption.

The current problems are sufficient to merit a significant review, and this review may require a whole new assessment of the phylogenetic analyses if the authors have indeed forgotten about selecting outgroups and rooting the tree. Hopefully, and most likely, the authors can justify their root choice and name the outgroup in the paper.

If re-analyses of phylogenetic methods are required, that would constitute a major review. However, I am optimistic, and I am recommending to accept the after minor revision.

No comments (I have not reviewed the language again because I did it the first time, and I do not have the bandwidth to do that in redundancy).

Author Response

Comments and Suggestions for Authors

I commend the authors for improving the manuscript and responding to all of my original comments. However, some significant issues remain. Specifically, phylogenetic methodology is more clearly described, but some details are lacking.

We modified the manuscript according to the reviewer's suggestions and we clarified some points below.

For example:
1) Multiple alignment parameters are not detailed.

We have carefully read the bibliography to be sure of the level of details previously described and we also checked on the Bioedit software the parameters used for the alignment [1-11], and we think we have provided as much detail as possible about the alignment parameters, as no additional information is available in the software (line 172-175).

“They were then aligned using the Clustal W multiple sequence alignment software inte-grated into BioEdit 7.2.5 (Manchester, UK). The best-fit nucleotide substitution pattern was determined using MEGA 7.0.26 [35] according to the corrected Akaike information criteri-on (AICc) [36].”

  1. Calvez, E.; Pommelet, V.; Somlor, S.; Pompon, J.; Viengphouthong, S.; Bounmany, P.; Chindavong, T.A.; Xaybounsou, T.; Prasayasith, P.; Keosenhom, S.; et al. Trends of the Dengue Serotype-4 Circulation with Epidemiological, Phylogenetic, and Entomological Insights in Lao PDR between 2015 and 2019. Pathogens 2020, 9, 728, doi:10.3390/pathogens9090728.
  2. Ngwe Tun, M.M.; Pandey, K.; Nabeshima, T.; Kyaw, A.K.; Adhikari, M.; Raini, S.K.; Inoue, S.; Dumre, S.P.; Pandey, B.D.; Morita, K. An Outbreak of Dengue Virus Serotype 2 Cosmopolitan Genotype in Nepal, 2017. Viruses 2021, 13, 1444, doi:10.3390/v13081444.
  3. Márquez, S.; Lee, G.; Gutiérrez, B.; Bennett, S.; Coloma, J.; Eisenberg, J.N.S.; Trueba, G. Phylogenetic Analysis of Transmission Dynamics of Dengue in Large and Small Population Centers, Northern Ecuador. Emerg Infect Dis 2023, 29, 888–897, doi:10.3201/eid2905.221226.
  4. Tun, M.M.N.; Kyaw, A.K.; Nabeshima, T.; Soe, A.M.; Nwe, K.M.; Htet, K.K.K.; Aung, T.H.; Htwe, T.T.; Aung, T.; Myaing, S.S.; et al. Detection of genotype-1 of dengue virus serotype 3 for the first time and complete genome analysis of dengue viruses during the 2018 epidemic in Mandalay, Upper Myanmar. PLOS ONE 2021, 16, e0251314, doi:10.1371/journal.pone.0251314.
  5. Hamel R, Surasombatpattana P, Wichit S, Dauvé A, Donato C, Pompon J, Vijaykrishna D, Liegeois F, Vargas RM, Luplertlop N, Missé D. Phylogenetic analysis revealed the co-circulation of four dengue virus serotypes in Southern Thailand. PLoS One. 2019 Aug 15;14(8):e0221179. doi: 10.1371/journal.pone.0221179. PMID: 31415663; PMCID: PMC6695175.
  6. Castonguay-Vanier J, Klitting R, Sengvilaipaseuth O, Piorkowski G, Baronti C, Sibounheuang B, Vongsouvath M, Chanthongthip A, Thongpaseuth S, Mayxay M, Phommasone K, Douangdala P, Inthalath S, Souvannasing P, Newton PN, de Lamballerie X, Dubot-Pérès A. Molecular epidemiology of dengue viruses in three provinces of Lao PDR, 2006-2010. PLoS Negl Trop Dis. 2018 Jan 29;12(1):e0006203. doi: 10.1371/journal.pntd.0006203. PMID: 29377886; PMCID: PMC5805359.
  7. Inizan C, O'Connor O, Worwor G, Cabemaiwai T, Grignon JC, Girault D, Minier M, Prot M, Ballan V, Pakoa GJ, Grangeon JP, Guyant P, Lepers C, Faktaufon D, Sahukhan A, Merilles OE Jr, Gourinat AC, Simon-Lorière E, Dupont-Rouzeyrol M. Molecular Characterization of Dengue Type 2 Outbreak in Pacific Islands Countries and Territories, 2017-2020. Viruses. 2020 Sep 25;12(10):1081. doi: 10.3390/v12101081. PMID: 32992973; PMCID: PMC7601490.
  8. Dupont-Rouzeyrol, M., Aubry, M., O’Connor, O. et al. Epidemiological and molecular features of dengue virus type-1 in New Caledonia, South Pacific, 2001–2013. Virol J 11, 61 (2014). https://doi.org/10.1186/1743-422X-11-61
  9. Aubry M, Roche C, Dupont-Rouzeyrol M, Aaskov J, Viallon J, Marfel M, Lalita P, Elbourne-Duituturaga S, Chanteau S, Musso D, Pavlin BI, Harrison D, Kool JL, Cao-Lormeau VM. Use of serum and blood samples on filter paper to improve the surveillance of Dengue in Pacific Island Countries. J Clin Virol. 2012 Sep;55(1):23-9. Doi: 10.1016/j.jcv.2012.05.010. Epub 2012 Jun 12. PMID: 22695001.
  10. Carrillo-Hernandez MY, Ruiz-Saenz J, Jaimes-Villamizar L, Robledo-Restrepo SM, Martinez-Gutierrez M. Phylogenetic and evolutionary analysis of dengue virus serotypes circulating at the Colombian-Venezuelan border during 2015-2016 and 2018-2019. PLoS One. 2021 May 28;16(5):e0252379. Doi: 10.1371/journal.pone.0252379. PMID: 34048474; PMCID: PMC8162668.
  11. Mendez, J.A., Usme-Ciro, J.A., Domingo, C. et al. Phylogenetic history demonstrates two different lineages of dengue type 1 virus in Colombia. Virol J 7, 226 (2010). https://doi.org/10.1186/1743-422X-7-226

2) Also, there is no indication in the paper about the outgroup for the phylogenetic analyses and how the root was selected. Outgroups are required for phylogenetic analysis, and rooting is needed to determine clades and character transformation's direction.

As mentioned in the review, the different trees are indeed rooted, but with the other DENV genotypes within the same serotype. Because of dengue virus genetic diversity, genotypes from the same serotype are commonly used for dengue virus phylogeny [12-20], otherwise resolution regarding close-related strains (which is the goal of our study) is lost.

According to remarks, we made Neighbor-Joining trees for each serotype to provide a quick response to the reviewer and to illustrate our concern regarding this point. These Neighbor-Joining phylogenetic trees were rooted with other DENV serotypes (example DENV-1 tree was rooted with DENV-3) as requested by the reviewer. We observed a decrease on the quality of our trees (branch delimitation is not visible, relation between the sequences obtained in the study is not well seen) with the addition of another serotype as outgroup. In the context of our study, which investigated dengue circulation in Guadeloupe and Martinique, we think it is preferable to keep the phylogenetic trees designed only with the genotype of each serotype as an outgroup for keeping their reading quality. This methodology has been extensively used in available literature (1-8) and we are sure of its adequacy and robustness.

However, if reviewer still maintains its decision, and his point is supported by the editor, we can eventually run the phylogenetic trees with the serotype as an outgroup with Maximum Likelihood method, and add them as supplementary data, but not in the manuscript main text. For this, we will need a minimum of seven days to provide the new tress as these analyses require time in our computers.

  1. Carrillo-Hernandez, M.Y.; Ruiz-Saenz, J.; Jaimes-Villamizar, L.; Robledo-Restrepo, S.M.; Martinez-Gutierrez, M. Phylogenetic and evolutionary analysis of dengue virus serotypes circulating at the Colombian–Venezuelan border during 2015–2016 and 2018–2019. PLOS ONE 2021, 16, e0252379, doi:10.1371/journal.pone.0252379.
  2. Dussart, P.; Lavergne, A.; Lagathu, G.; Lacoste, V.; Martial, J.; Morvan, J.; Cesaire, R. Reemergence of Dengue Virus Type 4, French Antilles and French Guiana, 2004–2005. Emerg Infect Dis 2006, 12, 1748–1751, doi:10.3201/eid1211.060339.
  3. Eldigail, M.H.; Abubaker, H.A.; Khalid, F.A.; Abdallah, T.M.; Musa, H.H.; Ahmed, M.E.; Adam, G.K.; Elbashir, M.I.; Aradaib, I.E. Association of Genotype III of Dengue Virus Serotype 3 with Disease Outbreak in Eastern Sudan, 2019. Virol J 2020, 17, 118, doi:10.1186/s12985-020-01389-9.
  4. Cao-Lormeau, V.-M.; Roche, C.; Aubry, M.; Teissier, A.; Lastere, S.; Daudens, E.; Mallet, H.-P.; Musso, D.; Aaskov, J. Recent Emergence of Dengue Virus Serotype 4 in French Polynesia Results from Multiple Introductions from Other South Pacific Islands. PLoS One 2011, 6, e29555, doi:10.1371/journal.pone.0029555.
  5. Chen R, Vasilakis N. Dengue--quo tu et quo vadis? Viruses. 2011 Sep;3(9):1562-608. doi: 10.3390/v3091562. Epub 2011 Sep 1. PMID: 21994796; PMCID: PMC3187692.
  6. Inizan, Catherine, Olivia O’Connor, George Worwor, Talica Cabemaiwai, Jean-Claude Grignon, Dominique Girault, Marine Minier, Matthieu Prot, Valentine Ballan, George Junior Pakoa, and et al. 2020. "Molecular Characterization of Dengue Type 2 Outbreak in Pacific Islands Countries and Territories, 2017–2020" Viruses 12, no. 10: 1081. https://doi.org/10.3390/v12101081
  7. Yenamandra, S.P., Koo, C., Chiang, S. et al. Evolution, heterogeneity and global dispersal of cosmopolitan genotype of Dengue virus type 2. Sci Rep 11, 13496 (2021). https://doi.org/10.1038/s41598-021-92783-y
  8. Shi, Y., Li, S., Li, X. et al. Epidemiological and molecular characterization of dengue viruses imported into Guangzhou during 2009–2013. SpringerPlus 5, 1635 (2016). https://doi.org/10.1186/s40064-016-3257-3
  9. Huang JH, Su CL, Yang CF, Liao TL, Hsu TC, Chang SF, Lin CC, Shu PY. Molecular characterization and phylogenetic analysis of dengue viruses imported into Taiwan during 2008-2010. Am J Trop Med Hyg. 2012 Aug;87(2):349-58. doi: 10.4269/ajtmh.2012.11-0666. PMID: 22855770; PMCID: PMC3414576.

3) Finally, on the images for the trees, the caption does not wholly describe the image. For example, the values on the branches are not named in the caption.

We added in the corresponding figure captions a mention of bootstraps: “Bootstrap values are shown at the base of branch nodes.”

Reviewer 2 Report

The manuscript was modified following the reviewers' suggestions, therefore it can be accepted in the present form.

Minor English editing required

Author Response

Comments and Suggestions for Authors

The manuscript was modified following the reviewers' suggestions, therefore it can be accepted in the present form.

We would like to thank the reviewer for the comments on the manuscript.